# OpenReview forum: "Certifying Robustness of Agent Tool-Selection Under Adversarial Attacks"
_ICLR.cc/2026/Conference — Submitted to ICLR 2026_

### Official Review · Reviewer_dCqm · 2025-10-26

**Soundness:** 3
**Presentation:** 3
**Contribution:** 3
**Rating:** 6
**Confidence:** 4

**Summary:**

The paper studies robustness of LLM agent tool selection, the two‑stage process in which a retriever surfaces a top‑N slate of tools and the agent chooses one to execute. It introduces CATS (Certification of Agentic Tool Selection), a statistical framework that treats each adversarial interaction as a Bernoulli trial and computes a high‑confidence lower bound on robust accuracy via Clopper–Pearson intervals. The attacker can inject up to k tools and adapt across R rounds using feedback from the agent’s previous choices; the refinement is modeled as a Markov process over tool metadata. Experiments use BFCL single‑tool tasks with M=300 tools, top‑N=10 retrieval (MiniLM-L6-v2), and several models as selectors (Llama‑3.1‑8B, Gemma‑3‑4B, Mistral‑7B, Phi‑4‑14B; Gemini‑2.5 Flash appears as an attacker). The paper defines five attack families (Adversarial Selection, Top‑N Saturation, Privilege Escalation, Abstention Trigger, Intent Shifting) and reports that the certified bound collapses toward zero under strong adaptive attacks (e.g., R=10), even when clean accuracy is high. A causal ablation shows low robustness (<0.5) even with forced inclusion of the correct tool in the slate, implicating both retriever and selector (Figure 3, p.8). The work argues CATS is the first formal statistical certification tailored to discrete tool selection rather than continuous perturbations or output text.

**Strengths:**

- First certification framework tailored to discrete tool selection with iterative adaptive attacks; clean formalization of pipeline and adversary space.
- Comprehensive experiments across models and attacks; informative ablations on rounds, budgets, retrievers, and frameworks
- Clear motivation (Figure 1, p.2), self‑contained algorithms (App. A.2), and compact visual summaries (Figure 2, p.8; Figures 4–8, pp.18–20).
- Reveals large robustness gaps between clean and certified performance; provides a reusable evaluation harness that can guide defense development for agentic systems.

**Weaknesses:**

- The lower bound is computed against the authors’ ∆adv class (templated, LLM‑generated, Markovian refinement). Claims of “worst‑case performance” should be qualified; results do not imply bounds over all possible adversaries or non‑Markov strategies. Provide an explicit statement of scope in §3.6
- Retrieval uses a single embedding model with Top‑N=10; more realistic settings include Unicode normalization, near‑duplicate clustering, homoglyph canonicalization, and slate diversification/quotas. Including at least one defended retriever baseline would strengthen conclusions about systemic vulnerability.
- The paper assumes a privilege field and compares to a user budget, but the user privilege model and enforcement are not specified; clarify how πuser is set and judged in experiments
- Results are on BFCL single‑tool calls with synthetic narrative context. It would be valuable to test on real tool stores or MCP/OpenAPI‑derived corpora and to vary M and N systematically to show scaling trends.

**Questions:**

- How generalizable is this certification to other adversarial strategies (e.g., non-Markov or non-templated attacks)? Could you explicitly clarify this scope in §3.6?
- The experiments use a single embedding-based retriever with Top-N = 10. Have you considered evaluating more realistic retrieval settings, such as Unicode normalization, near-duplicate clustering, homoglyph canonicalization, or slate diversification, to simulate defended retrievers? Including one defended retriever baseline could strengthen the claim of systemic vulnerability.
- In the Privilege Escalation attack, the paper assumes a privilege field π(t) and compares it to a user privilege πuser. How is πuser defined and enforced in your experiments? Are privilege mismatches detected through metadata rules or simulated policy constraints?
- All evaluations are conducted on BFCL single-tool tasks with synthetic narrative context. Have you tested or do you plan to test CATS on real tool stores (e.g., MCP/OpenAPI-derived corpora) or vary M and N systematically to analyze scaling behavior and generalizability?

---

> ### Author Response · Authors · 2025-11-20
> **Response to Reviewer dCqm (1/2)**
>
> We thank the reviewer for the insightful feedback. The weaknesses and questions raised are constructive and have led to new experiments, presented below, that we believe strengthen the paper.
>
> > **1\. "The lower bound is computed against the authors' Aadv class... Claims of 'worst-case performance' should be qualified; results do not imply bounds over all possible adversaries or non-Markov strategies. Provide an explicit statement of scope in §3.6" / "How generalizable is this certification to other adversarial strategies (e.g., non-Markov or non-templated attacks)? Could you explicitly clarify this scope in §3.6?"**
>
> **Response:**  The certified bound is relative to the class of adversaries ($A\_{adv}$). We have explicitly added a statement in Section 3.4 to restate that our lower bound is certified relative to the class of Markovian Refinement Adversaries.
>
> We have also clarified that while the CATS framework itself (the n-trial Bernoulli sampling) is general and agnostic to the attack strategy, the specific bound we compute is a certified lower bound on performance *against this specific distribution*. As detailed in our response to Reviewer 182B, we define a potent adversarial distribution and provide statistical guarantees on the agent's performance against samples from this distribution.
>
> > **2\. "Retrieval uses a single embedding model with Top- $N=10$.. Including at least one defended retriever baseline would strengthen conclusions about systemic vulnerability." / "Have you considered evaluating more realistic retrieval settings, such as Unicode normalization, near-duplicate clustering, homoglyph canonicalization, or slate diversification, to simulate defended retrievers? Including one defended retriever baseline could strengthen the claim of systemic vulnerability."**
>
> **Response:**  We thank the reviewer for this suggestion. We have conducted two new experiments based on the defense baselines the reviewer has suggested. The first defense method is a **Defended Retriever** enhanced with de-duplication (to catch copied tools) and homoglyph canonicalization (to catch simple obfuscation) and the second defense method is an **Anomaly Detection Monitor** that flags tools containing suspicious keywords.
>
> The results (which can be found in the table in our Response to Reviewer 182B) show that while simple defenses restore robustness against simple attacks, they fail to improve robustness against sophisticated adaptive attacks. Neither defense provided statistically significant protection against Adversarial Selection (16% vs 18%/21%) or Intent Shifting (66% vs 68%). This new analysis has been added to the appendix as a call to action for further research into defense mechanisms for tool-calling agents.
>
> > **3\. "The paper assumes a privilege field and compares to a user budget, but the user privilege model and enforcement are not specified; clarify how $\\pi\_{user}$ is set and judged in experiments" / "How is $\\pi\_{user}$ defined and enforced in your experiments? Are privilege mismatches detected through metadata rules or simulated policy constraints?"**
>
> **Response:**  We appreciate the reviewer pointing this out. In our experiments, $\\Pi\_{user}$ was considered to be the general 'user' level. The 'Privilege Escalation' attack (defined in Sec 3.3.3, Appendix A.1) generated tools with `privilege_level: 'admin'`. A failure was recorded if the agent selected any tool with the 'admin' privilege even though it was given the knowledge that it did not have sufficient privileges to access the function and that it should only use tools it meets the privilege requirements for. We have added this explicit definition to Section 4.2.

---

> ### Author Response · Authors · 2025-11-20
> **Response to Reviewer dCqm (2/2)**
>
> > **4\. "Results are on BFCL single-tool calls with synthetic narrative context. It would be valuable to test on real tool stores or MCP/OpenAPI-derived corpora and to vary M and N systematically to show scaling trends." / "Have you tested or do you plan to test CATS on real tool stores (e.g., MCP/OpenAPI-derived corpora) or vary M and N systematically to analyze scaling behavior and generalizability?"**
>
> **Response:**  We thank the reviewer for this suggestion, and we agree that testing on larger, real-world corpora is a valuable validation step. We have now run this experiment, applying our CATS framework to a large-scale, industry-standard tool pool populated from the **OpenAPI Specification** \[4\]. The following Table has been added to Appendix B.9 as Table 8\.
>
> **Table: Certified Robust Accuracy (Lower Bound) on the OpenAPI Tool Pool ($M=300, N=10$)**
>
> | Attack Type | Robust Accuracy |
> | :---- | :---- |
> | **Adversarial Selection** | **15%** |
> | **Top-N Saturation** | **18%** |
> | Privilege Escalation | 45% |
> | Abstention Trigger | 62% |
> | Intent Shifting | 67% |
>
> Our findings confirm that the severe vulnerabilities we identified hold true in this new setting. Under these conditions and on Gemma-3 as the attacker and Llama3.1 as the defender model, certified robust accuracy still collapsed under the most challenging settings, reaching just 15% for Adversarial Selection and 18% for Top-N Saturation. Privilege Escalation also remained a significant threat, reducing robust accuracy to 45%. While the less severe attacks like Intent Shifting (67%) and Abstention Trigger (62%) showed more resilience, these results demonstrate that our core finding is directly applicable to practical, real-world scenarios.
>
> Furthermore, we have analyzed the scaling behavior as requested by varying both the slate size ($N$) and the total tool pool size ($M$). First, we held the tool pool constant ($M=300$) and varied the slate size ($N$). The results for the two most severe threats the agent struggled with are shown below, added in Appendix B.10 as Tables 9 and 10:
>
> **Table: Impact of Slate Size ($N$) on Robust Accuracy ($M=300$)**
>
> | N (Slate Size) | Top-N Saturation | Adversarial Selection |
> | :---- | :---- | :---- |
> | 5 | 5% | 19% |
> | **10 (Baseline)** | **18%** | **15%** |
> | 15 | 23% | 13% |
>
> Second, we held the slate size constant ($N=10$) and varied the total tool pool size ($M$).
>
> **Table: Impact of Pool Size ($M$) on Robust Accuracy ($N=10$)**
>
> | M (Pool Size) | Top-N Saturation | Adversarial Selection |
> | :---- | :---- | :---- |
> | 100 | 20% | 18% |
> | **300 (Baseline)** | **18%** | **15%** |
> | 500 | 17% | 12% |
>
> \[1\] [https://help.openai.com/en/articles/8798878-building-and-publishing-a-gpt](https://help.openai.com/en/articles/8798878-building-and-publishing-a-gpt)
>
> \[2\] [https://help.zapier.com/hc/en-us/articles/18755649454989-App-versions-in-Zapier](https://help.zapier.com/hc/en-us/articles/18755649454989-App-versions-in-Zapier)
>
> \[3\] [https://docs.langchain.com/oss/python/langchain/overview](https://docs.langchain.com/oss/python/langchain/overview)
>
> \[4\] [https://swagger.io/specification/](https://swagger.io/specification/)

---

> > ### Comment · Reviewer_dCqm · 2025-11-21
> > **Keep my score**
> >
> > Thanks for your reply. I do not have any further questions now. I think this is a solid paper that clearly deserves a score of 6; however, it still falls short of an 8 (this is unrelated to the authors' response), possibly due to limitations in depth. Therefore, I will keep my score at 6.
> >
> > In addition, I noticed that reviewer 4cnB gave a score of 2 and provided comments that appear quite harsh. I would suggest that reviewer 4cnB approach a paper from the perspective of identifying its strengths and why it may be acceptable, rather than focusing primarily on why it should be rejected.

---

> > > ### Author Response · Authors · 2025-11-26
> > >
> > > Thank you for your time in reviewing our rebuttal. We appreciate your feedback and your support of our work in the broader review process.
> > > Regarding the "limitations in depth", we would welcome any opportunity to deepen the contribution of our work, so if you have specific insights on which directions you feel would be most effective for this purpose or any other remaining reservations, please let us know.

---

### Official Review · Reviewer_mWWQ · 2025-10-26

**Soundness:** 4
**Presentation:** 4
**Contribution:** 3
**Rating:** 8
**Confidence:** 3

**Summary:**

The authors study the setting of adversarially attacking the tool-selection process for agentic LLMs. They devise a small collection of prompts which they use to get a model to generate attacks. The attacks focus on two areas: (1) the slate selection phase (choosing which tools to consider in an initial narrowing-down phase), and (2) tool selection (choosing which tool to use from the narrowed-down slate). They find that models are susceptible to both families of attack.

**Strengths:**

## originality
As far as I know, this is the first work to explicitly study the threat model of attacking the tool pool.

## quality
The paper is well-written and the experiments are compelling.

## clarity
The paper is well written and overall quite clear.

## significance
It's not obvious exactly how important a threat model this is, given that it assumes no steps are taken to curate or moderate the tool pool. However, it is still interesting and a worthwhile addition to the literature.

**Weaknesses:**

It would be nice if the authors could explain a bit more clearly why their threat model is realistic. Or, if it's not realistic, to acknowledge it or explain how it could still be a problem in specific situations. My default intuition is that most of these issues go away if there is moderation of the tool pool: if it's a private company, then all their tools will be internal; if it's a public setting, then I would assume there would be some maintainers as in open-source projects who would check for this kind of malicious tool. Would tool certification/curation just solve this problem?

I would also like more discussion of what other defenses would look like, beyond tool certification/curation.

It would be good if the authors could be more clear in the paper (top and middle of page 6) that their attacks are in fact simply different prompts given to an LLM as input (which I later understood by looking at the Appendix). It was not clear just from reading that page.

**Questions:**

## Questions

137: I would really like citations supporting the idea that these tools can be authored by anyone. This is an important point for your paper, since the importance of the setting hinges on this being true.
202: what is semantic manipulation? Could you please explain it?
247: regarding the adaptive update, how much better is this than a simple best-of-N attack approach?
291: are you allowing k >= N? It seems like you're not, but this should be made very clear.
313: how do you choose $r$ and $k$? These seem pretty important.
349: "stability" is a weird thing to say here. It's just "to reduce noise".
360: in the multi-turn setting, what is k?
375: for top-N Saturation, is k < N? It seems that it should be. However, "saturation" and "flooding" give the image of all the N tools in the slate being malicious. Maybe you can change the language here to make it more clear.
Figure 2: what happened to the gemma3 orange bars in the first two plots?
Figure 3: can you also simply report what proportion of the time the correct tool wasn't in the slate? that seems like a much simpler way of answering this question, and you've already done the experiments for it.

## Suggestions

line 26: "severe" feels a bit strong
line 48: please provide a justifying citation to support the notion that anyone can publish malicious tools.
line 65: citet -> citep
line 68: citet -> citep
87: agentic systems -> agentic tool-calling systems
138: same as previous comment
189: consider putting a \quad after the comma before the t
191: excludes -> contains no
192: misleading -> wrong
256: I don't really understand this sentence
top of page 6: I didn't understand what exactly these attacks were until I looked at the appendix
301: at this point, it wasn't obvious to me how Privilege Escalation is different from Adversarial Selection. I think explaining clearly that these are all just different prompts would help a lot.
354: the Gemma3 citation is messed up
379: "attacker model" this is literally the first time I understood there was an attacking model. Please make it clear earlier
Figure 2: please make all plots have the same x axis order
Figure 3: this is a table, not a figure
436: while -> While

---

> ### Author Response · Authors · 2025-11-20
> **Response to Reviewer mWWQ (1/2)**
>
> We thank the reviewer for their positive assessment and their exceptionally thorough and constructive feedback. We have incorporated them to strengthen the polish of our paper. We address the reviewer's concerns and questions below.
>
> > **1\. "It would be nice if the authors could explain a bit more clearly why their threat model is realistic... Would tool certification/curation just solve this problem?" / "137: I would really like citations supporting the idea that these tools can be authored by anyone. This is an important point for your paper, since the importance of the setting hinges on this being true."**
>
> **Response:**  We agree that showing how realistic the threat model is important for the points presented in our paper. We have revised the paper to be more explicit that our threat model is most potent for open ecosystems, such as the OpenAI GPT Store or the Zapier marketplace.
>
> We also add citations to substantiate this. For example, OpenAl's review process for the GPT Store \[1\] focuses on policy compliance and verifying domain ownership, not a comprehensive security audit of third-party API logic. An even clearer case is Zapier \[2\], which allows developers to push new versions after initial approval without a subsequent review, creating a "bait-and-switch" vulnerability. Open-source frameworks like LangChain \[3\] rely on standard pull-request reviews, which check for functional correctness but are not formal security audits.
>
> While private tool pools made up entirely of in-house tools are safer compared to systems that compile tools from open sources, it simultaneously negates the primary advantage of agentic systems: extensibility. The ability to access a dynamic ecosystem of third-party tools (such as the ones in OpenAI GPT Store or Zapier Marketplace) to solve new tasks. Enforcing a closed, in-house-only ecosystem restricts its ability to interact with the broader pool of tools.
>
> In addition, to empirically validate that this threat is practical and not just theoretical, we ran new experiments (per reviewer dCqm's suggestion) on a large-scale, industry-standard tool pool derived from the **OpenAPI Specification** \[4\]. The findings confirm that the severe vulnerabilities we identified hold true when generalizing from the academic benchmark BFCL to industry-standard tool pools. Certified robust accuracy still collapsed in the most challenging adversarial settings (e.g., 15% for Adversarial Selection), a result nearly identical to our  baseline BFCL experiments, showing the generalizability of our core analyses in real-world scenarios. These results can be found in Appendix B.9, Table 8\.
>
> **Table: Certified Robust Accuracy (Lower Bound) on the OpenAPI Tool Pool ($M=300, N=10$)**
>
> | Attack Type | Robust Accuracy |
> | :---- | :---- |
> | **Adversarial Selection** | **15%** |
> | **Top-N Saturation** | **18%** |
> | Privilege Escalation | 45% |
> | Abstention Trigger | 62% |
> | Intent Shifting | 67% |
>
> > **2\. "I would also like more discussion of what other defenses would look like, beyond tool certification/curation."**
>
> **Response:**  We have added more discussion about potential defenses in Section 7\. We position CATS as the necessary framework to evaluate future solutions (such as curation, defended retrievers, or metadata sanitization). Furthermore, we direct the reviewer to our response to the feedback from Reviewer 182B (Point \#2), where we have run new experiments on two such defenses (a Defended Retriever with de-duplication and a metadata Anomaly Detection monitor). We leave the experiment's results below for your convenience.
>
> The additional experiments showed that while these defenses can increase robustness against simple attacks (like Top-N Saturation), they are insufficient against our adaptive semantic attacks (like Adversarial Selection).
>
> **Table: Certified Robust Accuracy (Lower Bound) under Baseline Defenses**
>
> | Attack Type | Adaptive Strategy (Ours) | Defense 1: Defended Retriever | Defense 2: Anomaly Detection |
> | :---- | :---- | :---- | :---- |
> | **Adversarial Selection** | **16%** | **18%** | **21%** |
> | **Top N Saturation** | 18% | 42% | 20% |
> | **Privilege Escalation** | 60% | 60% | 77% |
> | **Abstention Trigger** | 62% | 62% | 82% |
> | **Intent Shifting** | 66% | 66% | 68% |
>
> > **3\. "It would be good if the authors could be more clear in the paper (top and middle of page 6\) that their attacks are in fact simply different prompts given to an LLM as input..."**
>
> **Response:**  We appreciate the suggestion to improve the clarity and agree with the suggestion. We have revised Section 3.3.2 to explicitly state upfront that our adversary is "modeled as an LLM agent that uses a structured prompting strategy" and immediately refer to Appendix A.1 for the exact templates.

---

> ### Author Response · Authors · 2025-11-20
> **Response to Reviewer mWWQ (2/2)**
>
> > **4\. "202: what is semantic manipulation? Could you please explain it?"**
>
> **Response:**  We define 'semantic manipulation' as exploiting the LLM's interpretation of text, rather than a technical flaw. Our 'Persuasion Cues' (Sec 3.3.2), such as adding 'v2' or 'official', are examples of this manipulation in the sense that they don't change the tool's function but semantically persuade the LLM selector that the tool is more desirable. We have clarified this term in the revised paper.
>
> > **5\. "247: regarding the adaptive update, how much better is this than a simple best-of-N attack approach?"**
>
> **Response:**  As per the reviewer's request, we have conducted an ablation study comparing our stateful, adaptive adversary against a non-adaptive, 'Best-of-N' (BoN) baseline and have created the table below with the lower bound on the robust accuracy. The model pair used was Gemma-3 as the attacker model and Llama3.1 as the defender model so that the results can be directly compared against the results in Figure 2\. This table can be found in Appendix B.7, Table 6\.
>
> **Table: Adaptive Strategy vs. Best-of-N (BoN)**
>
> | Attack Type | Adaptive Strategy (Ours) | Best-of-N Strategy (Baseline) |
> | :---- | :---- | :---- |
> | **Adversarial Selection** | **16%** | **40%** |
> | **Top N Saturation** | 18% | 20% |
> | **Privilege Escalation** | 60% | 85% |
> | **Abstention Trigger** | 62% | 90% |
> | **Intent Shifting** | 66% | 92% |
>
> For simple, brute-force attacks (e.g., Top-N Saturation), which primarily target the retriever, the BoN strategy reduces robustness to a similar degree as our adaptive one (e.g., \~20% vs. 18% Robust Accuracy). However, for complex, semantic attacks (e.g., Privilege Escalation, Intent Shifting, and Abstention Trigger), the agent maintains significantly higher robustness against the BoN strategy.  Our experiments show that robust accuracy against BoN for these attacks remains high (e.g., 85-92%). Our adaptive attacker is the only method that significantly reduces robust accuracy to 60-66% for these complex attacks.
>
> > **6\. "291: are you allowing $k\>=N$? It seems like you're not, but this should be made very clear." / "360: in the multi-turn setting, what is k?" / "375: for top-N Saturation, is $k\<N?$ It seems that it should be. However, 'saturation' and 'flooding' give the image of all the N tools in the slate being malicious. Maybe you can change the language here to make it more clear."**
>
> **Response:**  Yes, we assume $k \< N$. We have made this assumption explicit in Sec 3.2, as if $k \\ge N$, the Top-N Saturation attack becomes trivial. We also clarify the language as follows: "Saturation" refers to flooding the retriever's similarity scores to push the correct tool out, not necessarily filling all $N$ slots. For the multi-turn setting, the budget $k$ is fixed. At each round $r$ of the adaptive attack, the adversary's new set of $k$ tools replaces the $k$ tools from round $r-1$. The total number of malicious tools in the pool at any time is always $k$.
>
> > **7\. "313: how do you choose k and R? These seem pretty important."**
>
> **Response:**  We analyze these as key hyperparameters of the threat model. We provide a full ablation on the effect of $R$ (rounds) in Table 1 (Appendix B.1) and an ablation on the effect of $k$ (budget) in Table 4 (Appendix B.4), showing robustness degrades as either increases. We chose $R=10$ based on preliminary experiments showing that robustness degradation saturates around this point, and $k=5$ (where $N=10$) to simulate a minority injection.
>
> > **8\. "349: 'stability' is a weird thing to say here. It's just 'to reduce noise'."**
>
> **Response:**  We thank the reviewer for pointing out this phrasing improvement. We have changed this wording from 'To ensure stability' to 'to reduce variance'.
>
> > **9\. "Figure 2: what happened to the gemma3 orange bars in the first two plots?"**
>
> **Response:**  The bars are not missing; the certified robust accuracy for Gemma-3 in those cases were effectively zero ($\<=0.01$), leading to the bar being difficult to see. We have added a note to the caption to make this clear.
>
> > **10\. "Figure 3: can you also simply report what proportion of the time the correct tool wasn't in the slate? that seems like a much simpler way of answering this question, and you've already done the experiments for it."**
>
> **Response:**  We thank the reviewer for this clarification. The data shows the correct tool was missing from the slate in 21% of trials at 1 round, 62% at 5 rounds, and 89% at 10 rounds. We have added this clarification in Section 5\.

---

> ### Author Response · Authors · 2025-11-26
>
> Thank you again for your thorough review and thoughtful questions. We hope our responses in the rebuttal have fully addressed your concerns regarding additional defense baselines, the realism of the threat model, and the clarity of the CATS framework. If there are any remaining questions or further concerns, please let us know, we would be happy to clarify or provide additional details. We appreciate your time and consideration throughout the review process.

---

### Official Review · Reviewer_4cnB · 2025-10-30

**Soundness:** 2
**Presentation:** 2
**Contribution:** 2
**Rating:** 2
**Confidence:** 4

**Summary:**

This paper presents CATS, a statistical framework for verifying the robustness of agent tool selection. CATS models tool selection as a Bernoulli process. By simulating multiple rounds of adaptive attacks (where attackers can iteratively optimize malicious tools based on the agent's historical selection), it generates a high-confidence lower limit of accuracy, thereby quantifying the agent's performance in the worst scenario. Experiments show that under multiple rounds of attacks, the authentication robustness of multiple SOTA LLM agents drops sharply to nearly zero, revealing serious security threats in the tool selection process.

**Strengths:**

+ The issue is of practical significance: With the widespread application of LLM agents in tool invocation, the robustness of tool selection is indeed a critical and understudied security problem.
+ The experimental system is rich: multiple models and various attack types were evaluated, and the vulnerabilities of the retrieval device and the selector were deeply analyzed through ablation experiments.

**Weaknesses:**

- The key points of the paper and the content of the method section are out of balance: The core content of this paper is the robustness evaluation framework, but the methods section devotes a considerable amount of space to detailing the classification and implementation of attack methods (such as Top-N Saturation, Abstention Trigger, etc.). This leads readers to feel confused when understanding the core mechanisms of the certification framework itself (such as the composition of multiple rounds of experiments, the definition of the Bernoulli process, and the calculation of confidence intervals), and they are not clear about the priorities.
- Key method details are missing: Although the paper presents various attack types, it does not elaborate on how these attacks are specifically implemented in the system. For example, how are the three attack types such as Top-N Saturation and Abstention Trigger implemented?
- The lack of a clear threat model: The paper does not explicitly define the attacker's specific capabilities, knowledge boundaries, and restrictive conditions (for example, whether the attacker can access the retrieval device, whether they can modify existing tools, etc.), which casts doubt on the rationality and universality of the attack scenario.

**Questions:**

- What is the core mechanism of the certification framework?
- What are the specific Settings of the threat model?

---

> ### Author Response · Authors · 2025-11-20
> **Response to Reviewer 4cnB**
>
> We thank the reviewer for their detailed feedback. The reviewer's main concerns appear to be about clarity and presentation (i.e. imbalance of the methods section, missing threat model). We have addressed each concern below and have revised the paper to make each clarification more explicit.
>
> > **1\. "The key points of the paper and the content of the method section are out of balance... This leads readers to feel confused when understanding the core mechanisms of the certification framework itself..." / "What is the core mechanism of the certification framework?"**
>
> **Response:**  We thank the reviewer for this feedback regarding the presentation structure. To improve clarity, we have revised Section 3 to explicitly distinguish the problem definition from the solution. We have clearly framed the attack families (Sec 3.3.3) as the potential threats that our framework is designed to evaluate, and then introduced Section 3.4 as the "core certification mechanism" for quantifying robustness within that space.
>
> > **2\. "Key method details are missing... how are the three attack types such as Top-N Saturation and Abstention Trigger implemented?"**
>
> **Response:**  We thank the reviewer for this question. The high-level definitions can be found in Section 3.3.3, and the general implementation strategy is in Section 3.3.2 and expanded further upon in Appendix A.1, described as "a structured prompting strategy with an attacker LLM. The core template instructs the model to create a tool with metadata that is semantically similar to a target tool but functionally distinct. For iterative refinement, the prompt is augmented with the defender agent's previous choice and the outcome, guiding the attacker LLM to generate a more persuasive variant."
>
> For the attack specific descriptions, Top-N Saturation is defined in Sec 3.3.3: "targets the retriever by using deceptive similarity techniques to inject k near-duplicate variants..." and Abstention Trigger is defined in Sec 3.5: "embedding refusal-inducing textual content into a tool's metadata..." The exact prompt templates can be found in Appendix A.1. We have made the reference to these descriptions more explicit in the main paper.
>
> > **3\. "The lack of a clear threat model: The paper does not explicitly define the attacker's specific capabilities, knowledge boundaries, and restrictive conditions (for example, whether the attacker can access the retrieval device, whether they can modify existing tools, etc.), which casts doubt on the rationality and universality of the attack scenario." / "What are the specific Settings of the threat model?"**
>
> **Response:**  We thank the reviewer for this point, as it allows us to clarify our threat model. We explicitly define the threat model in Section 3.2, "Adversary Model and Access," where we lay out the attacker's capabilities and their restrictions.
>
> We state that the attacker "cannot modify the original tool pool $T$, the retrieval function $s$, or the agent $L$". The attacker's only capability is "to inject new tools into the repository up to a fixed budget of $k$". We chose this specific, constrained model because it is the most realistic and practical threat, as it mirrors real-world platforms like the **OpenAI GPT Store** \[1\] or **Zapier Marketplace** \[2\], where the barrier to publishing a tool is low, but the barrier to compromising model weights is high. We also define the attacker's knowledge as the realistic ability to observe or probe the system to align their malicious tools with user intents, which is necessary for their attacks to be surfaced. This is a standard "black-box" access model available to users of public agentic systems. We have made these real-world parallels more explicit in Section 3.2.
>
> \[1\] [https://help.openai.com/en/articles/8798878-building-and-publishing-a-gpt](https://help.openai.com/en/articles/8798878-building-and-publishing-a-gpt)
>
> \[2\] [https://help.zapier.com/hc/en-us/articles/18755649454989-App-versions-in-Zapier](https://help.zapier.com/hc/en-us/articles/18755649454989-App-versions-in-Zapier)

---

> ### Author Response · Authors · 2025-11-26
>
> Thank you again for your thorough review and thoughtful questions. We hope our responses in the rebuttal and the revisions to the manuscript have fully addressed your concerns regarding the clarity of the threat and certification model. If there are any remaining questions or further concerns, please let us know, we would be happy to clarify or provide additional details. We appreciate your time and consideration throughout the review process.

---

### Official Review · Reviewer_182B · 2025-11-03

**Soundness:** 2
**Presentation:** 2
**Contribution:** 2
**Rating:** 2
**Confidence:** 3

**Summary:**

This paper studies adversarial attack on the tool-selection stage of agentic systems. Specifically, the attacker could inject malicious tools and mislead agents to select them.  It formalizes robustness as the probability that the agent still picks an intent-satisfying tool even when an adaptive adversary can inject up to $k$ malicious tools and iteratively refine them over $R$ rounds. The proposed framework, CATS (Certification of Agentic Tool Selection), treats each full multi-round interaction as a Bernoulli trial and uses Clopper–Pearson intervals to produce a high-confidence lower bound on “robust accuracy.” In particular, to study the worst case setting, the paper introduces an adaptive attacker that can dynamiclly refine its attacking policy based on the agent's previus choices. Experiment results show that under multi-round attacks the certified lower bound can collapse to near zero, and even with forced inclusion of the correct tool, certified robustness stays <50%, indicating both retrieval and selection are vulnerable

**Strengths:**

1. Studying adversarial attack on the tool-selection stage of agentic systems is well motivated.

2. Formalizing a multi-round problem instead of a single round is more realistic, which unlocks the potential to study advanced red team strategies (e.g. adaptive attacks studied in this paper).

3. Results are evaluated across multiple attack strategies (Top-N Saturation, Adversarial Selection, Privilege Escalation). The near-zero certified lower bound convincingly show that this is a real problem.

**Weaknesses:**

1. overclaim on novelty. As far as I know, this is not the **first** paper studies adversarial attack on the tool-selection stage. see https://arxiv.org/pdf/2412.10198 and https://arxiv.org/pdf/2508.02110v1.

2. Lack of experimenting with more defense methods. For example. how easy it is to catch these injected malicious tools? Can the blue team easiy select them with an additional monitor before using the retriever and selector?

3. Studying the worst-case setting of adaptive attacks is reasonable. However, in practice, the attacker might not really have access to the agents' detailed trajectories since they are often hided by companies?

**Questions:**

1. What was n (number of trials) per model/attack in practice? How sensitive were your lower bounds to halving n? A small table of “trials → CI width” would make the “certified” claim more concrete.

2. The current paper tests defender LLaMA-3.1 vs. attacker Gemma-3 as a “representative” strong attacker (P7). Did you try mismatched or weaker attackers? Do we still see near-zero bounds when the attacker LLM is strictly smaller or older than the defender?

3. The current paper focuses on single-tool tasks. How does the proposed attack adapt to multi-tool tasks?

---

> ### Author Response · Authors · 2025-11-20
> **Response to Reviewer 182B (1/3)**
>
> We thank the reviewer for their valuable suggestions and feedback. We have conducted new experiments based on the reviewer's feedback, which we believe strengthen the paper. We address the reviewer's concerns and questions below.
>
> > **1\. "As far as I know, this is not the first paper studies adversarial attack on the tool-selection stage."**
>
> **Response:** We thank the reviewer for these highly relevant citations, which are examples of the exact threat our framework is designed to measure. 'From Allies to Adversaries: Manipulating LLM Tool-Calling through Adversarial Injection' was a paper we used as an example to show the rising prevalence of attacks targeting the tool selection stage.
>
> However, the distinction we would like to highlight is that existing papers propose attacks, whereas our novelty lies in the **certification**, not the attack. As reviewers R-mWWQ and R-dCqm have noted in their respective reviews, our work is the 'first certification framework tailored to discrete tool selection' (R-dCqm).
>
> Our work addresses the limitation of existing evaluations, the combinatorial aspect of tool selection space. This space is defined by the joint distribution of user intents $u$ and the wide configuration of tools and their natural language metadata $t$ in the tool pool. Standard benchmarks such as BFCL only evaluate tool use in limited, non-adversarial settings, overlooking the exponential complexity that arise from combining diverse user goals with the semantic tool descriptions. While recent adversarial works (as the ones the reviewer pointed out) have correctly identified this attack surface, they focus solely on the qualitative question of whether an attack exists within this space. However, in such an unbounded semantic space, finding a single failure mode is likely possible for any model. Thus, an “existence proof” makes meaningful comparison between models impossible as all models would become equally “non-robust" if even a single attack is found.
>
> CATS addresses this by providing guarantees with respect to probabilistic distributions defined over this large space of adversarial tool definitions. Instead of searching for a qualitative answer, we answer a quantitative question of computing the probability that the model remains robust against randomly sampled adversarial tools. This can be used to bound the number of points for which the agent is robust. For example, if the distribution contains 100 billion points (whereas our actual space is functionally infinite) and we get a lower bound on robustness to be 20% with 95% confidence, then we can say with high confidence that the model is robust for 20 billion samples at least (while 80 billion points can be attacked), a quantitative guarantee that qualitative baselines are unable to provide. Since computing this exact probability would be infeasible for black-box models, we focus on providing the statistical guarantees.
>
> We contribute a method to construct a practical adversarial distribution and collect independent and identically distributed (I.I.D.) samples from it. This produces a statistically guaranteed lower bound on the agent's robustness in regards to this distribution, shifting the evaluation from the qualitative existence of an attack to a quantitative probability that the model remains robust against an adaptive threat distribution.

---

> ### Author Response · Authors · 2025-11-20
> **Response to Reviewer 182B (2/3)**
>
> > **2\. "Lack of experimenting with more defense methods. For example. how easy it is to catch these injected malicious tools? Can the blue team easiy select them with an additional monitor before using the retriever and selector?"**
>
> **Response:**  We agree this is an important area for future work. We frame CATS as the necessary prerequisite for developing such defenses, as we believe we cannot build a robust defense without a proper method to quantify that robustness.
>
> Below, we test two defenses suggested by reviewer dCqm: (1) a **Defended Retriever** (De-duplication \+ Homoglyph Canonicalization), which applies common techniques from information retrieval and text security \[1\]\[2\], and (2) **Anomaly Detection Monitoring**, a standard 'guardrail' technique used to filter malicious inputs trained on a separate split of the dataset disjoint from the evaluation set \[3\]. This table has been added to Appendix B.8 as Table 7\.
>
> **Table: Certified Robust Accuracy (Lower Bound) under Baseline Defenses**
>
> | Attack Type | Adaptive Strategy (Ours) | Defense 1: Defended Retriever | Defense 2: Anomaly Detection |
> | :---- | :---- | :---- | :---- |
> | **Adversarial Selection** | **16%** | **18%** | **21%** |
> | **Top N Saturation** | 18% | 42% | 20% |
> | **Privilege Escalation** | 60% | 60% | 77% |
> | **Abstention Trigger** | 62% | 62% | 82% |
> | **Intent Shifting** | 66% | 66% | 68% |
>
> The Defended Retriever's de-duplication logic was highly effective against Top-N Saturation, more than doubling the lower bound on robust accuracy from 18% to 42%. However, this retriever-level defense was insufficient against all other attack types, providing no significant benefit against Adversarial Selection (16% vs 18%) and all three semantic attacks.
>
> The Anomaly Monitor was effective against attacks that rely on simple, static keywords, increasing the lower bound on robustness for Privilege Escalation (60% to 77%) and Abstention Trigger (62% to 82%). However, it provided little benefit against our Adversarial Selection attack (16% vs 21%). Neither defense was effective in defending against Intent Shifting attacks (66% vs 68%).
>
> > **3\. "Studying the worst-case setting of adaptive attacks is reasonable. However, in practice, the attacker might not really have access to the agents' detailed trajectories since they are often hided by companies?"**
>
> **Response:**  We thank the reviewer for this important clarifying question. We have revised Section 3.3 to be more explicit. Our adaptive refinement process (Sec 3.3.1) does **not** assume access to private user trajectories. Instead, it models a realistic, offline search process where an attacker, acting as a regular user, can probe the public-facing agent.
>
> By observing which tool the agent selects in response to their own queries, they receive black-box feedback. They can then update their malicious tool's metadata and re-upload/re-probe, iterating $R$ times to discover the configuration that minimizes agent robustness. As detailed in our revised Section 3.2, this feedback loop observing the final selection is a realistic capability in open plugin ecosystems like the **OpenAI GPT Store**\[3\] or **Zapier Marketplace**\[4\], where an attacker can simply create an account and test the system.

---

> > ### Author Response · Authors · 2025-11-20
> > **Response to Reviewer 182B (3/3)**
> >
> > > **4\. "What was n (number of trials) per model/attack in practice? How sensitive were your lower bounds to halving n? A small table of 'trials \- CI width' would make the 'certified' claim more concrete."**
> >
> > **Response:**  We thank the reviewer for this suggestion. For our main experiments on the BFCL benchmark, we used $n=1000$ trials per model/attack. To address the topic of sensitivity, we performed the suggested analysis.
> >
> > The table below (which has been added as a visualization to Appendix B.6 as Figure 4\) shows the 95% Clopper-Pearson bounds for the most severe vulnerability (Adversarial Selection).
> >
> > **Table: Sensitivity Analysis of 95% CI Convergence (Adversarial Selection)**
> >
> > | Number of Trials ($n$) | Lower Bound | Upper Bound | Interval Width |
> > | :--- | :--- | :--- | :--- |
> > | 125 | 13.6% | 33.4% | 19.8% |
> > | 250 | 15.2% | 29.0% | 13.8% |
> > | 500 | 15.4% | 24.8% | 9.4% |
> > | **1000 (Ours)** | **16.0%** | **22.3%** | **6.3%** |
> > | 2000 | 16.4% | 20.9% | 4.5% |
> >
> > As shown, the confidence interval tightens significantly as $n$ increases. At $n=500$, the interval width widens to 9.4%, which introduces moderate uncertainty. At $n=1000$ (our choice), the width narrows to 6.3%, and increasing to $n=2000$ yields diminishing returns, reducing the width by only 2.0% while doubling the computational cost. This analysis confirms the expected statistical behavior and quantifies the trade-off between computational cost and bound precision.
> >
> > > **5\. "The current paper tests defender LLaMA-3.1 vs. attacker Gemma-3 as a 'representative' strong attacker (P7). Did you try mismatched or weaker attackers? Do we still see near-zero bounds when the attacker LLM is strictly smaller or older than the defender?"**
> >
> > **Response:**  We agree that showing generalizability across models of varying levels of power is important. Our main text (Figure 2\) only presented a single representative attacker (Gemma-3) for clarity and space. However, our full experiments include the mismatched and weaker attacker scenarios the reviewer requested. We direct the reviewer to Appendix C, Figures 5-9, which show the complete results for all 16 attacker-defender pairs (e.g., Llama-3.1 vs. Phi-4, Mistral vs. Gemma-3). This has been made more explicit in Section 5\.
> >
> > > **6\. "The current paper focuses on single-tool tasks. How does the proposed attack adapt to multi-tool tasks?"**
> >
> > **Response:**  We agree with the reviewer that this is an important direction for future work. We explicitly focused on single-tool tasks to specifically isolate the selection mechanism. Extending CATS to certify compositional robustness of multi-step plans is an important but non-trivial next step. In a multi-tool plan, every individual step relies on a correct previous selection from the tool pool. If the agent cannot robustly select a single tool $t$ from a slate $S$ under adversarial pressure, any complex plan built on that selection is compromised. By certifying the robustness of this single tool setting, our work provides the necessary lower-bound guarantee required before compositional robustness can even be meaningfully evaluated. We have added this to our conclusion as a key direction for future work.
> >
> > \[1\] Manku et al. "Detecting Near-Duplicates for Web Crawling" (2007).
> >
> > \[2\] Teja et al. “Modeling the Attack: Detecting AI-Generated Text by Quantifying Adversarial Perturbations” (2025)
> >
> > \[3\] Hawkins et al. “Machine Learning for Detection and Analysis of Novel LLM Jailbreaks” (2025)
> >
> > \[4\] [https://help.openai.com/en/articles/8798878-building-and-publishing-a-gpt](https://help.openai.com/en/articles/8798878-building-and-publishing-a-gpt)
> >
> > \[5\] [https://help.zapier.com/hc/en-us/articles/18755649454989-App-versions-in-Zapier](https://help.zapier.com/hc/en-us/articles/18755649454989-App-versions-in-Zapier)

---

> > > ### Author Response · Authors · 2025-11-26
> > >
> > > Thank you again for your thorough review and thoughtful questions. We hope our responses in the rebuttal have fully addressed your concerns regarding additional defense baselines, the realism of the threat model, and the novelty of the CATS framework. If there are any remaining questions or further concerns, please let us know, we would be happy to clarify or provide additional details. We appreciate your time and consideration throughout the review process.

---

### Author Response · Authors · 2025-11-20
**General Response to All Reviewers**

We thank the reviewers for their time and constructive feedback. To facilitate the review process, all revisions made to the manuscript in response to the feedback have been marked in **blue** text.

---

### Author Response · Authors · 2025-12-03
**Summary of Rebuttal and Key Revisions**

We provide a brief summary of the rebuttal process and the key revisions made to the manuscript. Reviewers mWWQ and dCqm have recognized the novelty of the certification framework and Reviewer dCqm has acknowledged that their concerns have been fully resolved following our response.

Besides minor clarifying questions on implementation details and additional ablation studies, the reviews focused on the following concerns that we have comprehensively addressed through new experiments and detailed clarifications:

> **Reviewer mWWQ**

**Concern:** Requested validation on larger tool pools and comparison against simpler "Best-of-N" baselines.

**Resolution:** We conducted new experiments using the OpenAPI Specification (an industry standard for agent tools), where robust accuracy collapsed to 15% just like our original experiments, validating the threat in real-world scenarios and the generalizability of our results to other data sources. We also added a Best-of-N ablation (Appendix B.7), showing that the adaptive scenario we evaluated is significantly more threatening than the BoN baseline.
> **Reviewer dCqm**

**Concern:** Requested evaluation of defense baselines (i.e., defended retrievers).

**Resolution:** We evaluated two defenses suggested by another reviewer: (1) A Defended Retriever (de-duplication + homoglyph canonicalization) and (2) Anomaly Detection Monitoring. While these improved robustness against naive attacks, they failed against the adaptive scenario, empirically validating the severity of the vulnerability and the need for the CATS certification framework to better reveal said vulnerability.
> **Reviewer 182B**

**Concern:** Questioned the novelty regarding the attack surface and requested defense experiments.

**Resolution:** We clarified that the primary contribution is the certification framework (quantifying worst-case robustness), not the existence of an attack. We also added the requested defense experiments (detailed above), which demonstrated that current defenses are insufficient and that formal certification is required.
> **Reviewer 4cnB**

**Concern:** Presentation balance between attack methods vs. certification framework, and clarity on the threat model.

**Resolution:** We restructured Section 3 to clearly distinguish the problem definition from the certification solution. We also explicitly defined the threat model (Section 3.2) to be analogous to open plugin ecosystems (i.e., OpenAI GPT Store), where attackers can publish tools but cannot access model weights. We have added relevant citations of  these real systems and platforms that follow the assumptions presented in our paper.

---

### Meta-Review · Area_Chair_dDX2 · 2026-01-07

**Summary:**

The paper proposes CATS, a statistical framework to certify the robustness of Large Language Model agents against adversarial attacks during the tool selection phase. The authors model the interaction as a Bernoulli process to derive high-confidence lower bounds on performance under adaptive attacks. Reviewers are diverging, with some appraising the novelty of applying formal certification to the discrete domain of tool selection, and others being more critical, raising concerns about the novelty of the attack surface, the realism of the threat model, and the lack of defense baselines.

**Reviewer Concerns:**

Addressed Concerns:

- Generalizability: The authors added experiments using the OpenAPI Specification tool pool, confirming that the vulnerabilities persist in real-world, industry-standard datasets.

Comparison to Baselines: A "Best-of-N" ablation study was added, demonstrating that the proposed adaptive attack is significantly more effective than standard baselines.

Outstanding Concerns:

- Realism of Threat Model: The authors clarified that the threat model reflects open ecosystem platforms (e.g., OpenAI GPT Store, Zapier), where attackers can publish tools without accessing model weights. But it is a still a bit different compared to popular general agent settings where tools are more internal things.

- Limited Novelty: Reviewer pointed out concurrent work exploring similar attacks. And the necessity for such certification over the attack is a not entirely convincing.

- Lack of Defenses: While the authors added baselines for defenses (defended retrievers, anomaly detection) that failed, the paper remains primarily a vulnerability report. The lack of a proposed, effective defense mechanism limits the practical utility of the work.

- Presentation and Balance: Reviewer 4cnB found the paper unbalanced, focusing heavily on attack details rather than the core certification mechanism.

**Reviewer Scores:**

currently 2268, unlikely to change in my opinion

---

### Decision · Program_Chairs · 2026-01-26

Reject